# Neurological Involvement in Children with COVID-19 and MIS-C: A Retrospective Study Conducted for More than Two Years in a Pediatric Hospital

**DOI:** 10.3390/children9121809

**Published:** 2022-11-24

**Authors:** Giulia Abbati, Federica Attaianese, Anna Rosati, Giuseppe Indolfi, Sandra Trapani

**Affiliations:** 1Paediatric Residency, Meyer Children’s University Hospital, Viale Pieraccini 24, 50139 Florence, Italy; 2Department of Health Sciences, University of Florence, Meyer Children’s University Hospital, Viale Pieraccini 24, 50139 Florence, Italy; 3Neurology Unit, Meyer Children’s University Hospital, Viale Pieraccini 24, 50139 Florence, Italy; 4NEUROFARBA Department, University of Florence, Meyer Children’s University Hospital, Viale Pieraccini 24, 50139 Florence, Italy

**Keywords:** neurological involvement, children, pediatric, SARS-CoV-2, COVID-19, MIS-C

## Abstract

This study aimed to evaluate the type and severity of neurological involvement in children with SARS-CoV-2 infection or multisystem inflammatory syndrome in children (MIS-C) and compare these findings between the two groups. Children hospitalized with the diagnosis of COVID-19 or MIS-C at Meyer Children’s Hospital between February 2020 and June 2022 were retrospectively studied. One hundred twenty-two patients were enrolled, 95 in the COVID-19 group and 27 in the MIS-C group. In the COVID-19 group, impairment of consciousness was found in 67.4% of patients, headache in 18.9% and about 16.8% of patients experienced seizures. In this group, three patients were diagnosed with arterial ischemic stroke and one patient was diagnosed with Guillain-Barré syndrome (GBS). In the MIS-C group, about 70% of patients experienced consciousness impairment, about 20% behavioral changes, and another 20% mood deflection. Neurological symptoms and signs were highly heterogeneous and could be differentiated in COVID-19 and MIS-C. Consciousness impairment remained the most frequent manifestation in both groups, potentially underlying an encephalopathy. We also highlight the importance of considering psychiatric symptoms in children with COVID-19 and/or MIS-C. Most neurological manifestations were mild in our series; however, severe complications such as ischemic stroke and GBS are worthy of note.

## 1. Introduction

Severe acute respiratory syndrome type 2 coronavirus (SARS-CoV-2) infection rapidly spread all over the globe from Wuhan, China, being declared a pandemic in February 2020 [1]. The infection can occur in children and adolescents mostly with a mild course, or even asymptomatically, but occasionally with a severe or even lethal evolution [2,3,4,5,6]. Coronavirus disease-2019 (COVID-19) may present not only with respiratory symptoms but also with various multisystemic involvement, including neurological impairment, even in children [7,8]. SARS-CoV-2, like other viruses of the same family SARS-CoV (2002) and Middle East respiratory syndrome coronavirus (MERS-CoV, 2013), can affect the nervous system, both central (CNS) and peripheral (PNS) [9]. Accordingly, the virus membrane receptor angiotensin-converting enzyme 2 (ACE2) is expressed even in the neuronal and glial cells [4].

Neurological involvement in SARS-CoV-2 infection was first described in adults in the first months of the pandemic, and an increasing number of neurological and psychiatric manifestations were reported in the literature [10,11,12,13,14,15]. Neurological involvement was described also in pediatric cases, and even neonatal cases, and several pediatric studies and series have been published to date, identifying an incidence ranging from 3.8% to 43% [5,16,17,18]. Neurological impairment has also been described during multisystem inflammatory syndrome in children (MIS-C) or pediatric inflammatory multisystem syndrome—temporarily associated with SARS-CoV-2 (PIMS-TS), being included in the diagnostic criteria, with a variable incidence from 11 to 67% [5,16,17,19,20,21,22,23,24,25]. Both in COVID-19 and MIS-C the neurological manifestations range in a wide spectrum, from mild and more frequent symptoms such as dysgeusia, headache, and mild transient impairment of consciousness, to severe manifestations such as seizures, ataxia, encephalitis, meningitis, acute ischemic stroke, demyelinating disorders, and coma. Occasionally, even fatal cases have been reported [5,6,16,17,18,22,24,26,27,28]. Psychiatric disorders have also been described, even in children [16,17,29].

We report the clinical presentation of a large cohort of children with COVID-19 and MIS-C in more than two years of the pandemic. This study aimed to evaluate the type and severity of neurological involvement during both COVID-19 and MIS-C, analysing demographic data, clinical manifestations, laboratory results, instrumental investigations, management, and outcome. A second aim was to compare these findings between the two groups, COVID-19 and MIS-C.

## 2. Materials and Methods

### 2.1. Study Design and Cases Definition

We retrospectively selected all children (age 0–18 years) admitted to pediatric wards of the Meyer Children’s University Hospital, from February 2020 to June 2022 with diagnosis of COVID-19 or MIS-C. Such patients were identified by specific discharge diagnoses according to the International Classification of Diseases, 9th Revision, Clinical Modification.

COVID-19 was diagnosed in symptomatic patients, identified by the positivity of polymerase chain reaction (PCR) for SARS-CoV-2 or an antigenic test on nasal swabs. MIS-C diagnosis was made based on Centers for Disease Control and Prevention (CDC) criteria: fever, laboratory evidence of inflammation, clinically severe illness requiring admission, with multisystem (≥2) organ involvement (cardiac, renal, respiratory, hematologic, gastrointestinal, dermatologic, or neurological), no alternative diagnoses and current or recent SARS-CoV-2 infection by PCR, antigen test or serology, or recent COVID-19 exposure [19].

Among patients with COVID-19 or MIS-C, we enrolled those with at least one neurological sign or symptom from mild to severe, and patients with neuropsychiatric disorders. Furthermore, in order to evaluate clinical differences depending on age, patients were divided into three groups (0–3, 3–10 and 10–18 years old). The whole cohort was divided into two groups: the COVID-19 group and the MIS-C group to evaluate differences.

From electronic medical records, we collected data about familial and medical histories, SARS-CoV-2 vaccination status, clinical manifestations (both neurological and non), blood tests (inflammatory markers, blood cells count, coagulation, cardiac function indexes), cerebrospinal fluid (CSF) exam, SARS-CoV-2 infection status (PCR/antigenic test or serology), instrumental investigations (electroencephalogram [EEG], electro-neuro-myography [ENMG], evoked potentials), CNS radiological examinations (computed tomography [CT]-scan, magnetic resonance imaging [MRI], magnetic resonance angiography [MRA], digital subtraction angiography [DSA]), management, disease course, and neurological outcome at discharge and follow-up.

Informed consent to study enrollment was acquired by parents or legal guardians at patient admission.

### 2.2. Statistical Analysis

Statistical analysis was performed using SPSS software (version 25-IBM Analytics). Categorical data were described as frequency and percentage. Non-normally distributed continuous variables were expressed as the median and interquartile range (IQR). The Pearson’s chi-squared test or Fisher exact test was used, where appropriate, to compare categorical variables, while the two-tailed Mann Whitney’s U Test was employed for the comparison between medians. A *p*-value < 0.05 was statistically significant.

## 3. Results

### 3.1. Epidemiology

Among 345 patients diagnosed with SARS-CoV-2 infection (*n* = 304) or MIS-C (*n* = 41), 122 (35.4%) were enrolled in the study, 95 (31.3%) in the COVID-19 group and 27 (65.9%) in the MIS-C group. The median age was 1.98 years (IQR 0.34–8.57). Male gender (*n* = 74, 60.7%) and Caucasian ethnicity (*n* = 101, 82.8%) were prevalent. The median age in the COVID-19 group (0.94 [IQR 0.21–6.98] years) was significantly lower than in the MIS-C group (8.50 [IQR 3.68–11.62] years). Twenty-four patients (19.7%) had a previous neurological or psychiatric disease, specifically febrile seizures (*n* = 6), headache (*n* = 5), tetraparesis and epilepsy (*n* = 3), language delay (*n* = 2), brain tumour (*n* = 2), and Arnold Chiari type I (*n* = 2). Other less frequent disorders were found in one patient each (Asperger syndrome, genetic epilepsy, sleep disorder, hydrocephalus, central apnoea, learning disability, cognitive delay, attention deficit hyperactivity disorder (ADHD), spina bifida, mood and eating disorders). Eleven patients (9%) received at least one dose of SARS-CoV-2 vaccination (three patients a complete cycle of three doses). Epidemiological, familiar and past medical history data are shown in Table 1.

### 3.2. Clinical Manifestations

Seventy patients (57.4%) presented neurological involvement at onset: 58 (61.1%) of the COVID-19 group and 12 (44.4%) of MIS-C one. Neurological signs and symptoms are summarized in Table 2 and Appendix A and classified by patients’ age in Figure 1. The extra-neurological manifestations are reported in Appendix A.

#### 3.2.1. COVID-19 Group

The most frequent neurological manifestations at onset were consciousness impairment (*n* = 34, 35%), headache (*n* = 12, 12.6%), and seizures (*n* = 10, 10.5%) (Appendix A). Considering the entire disease course, impairment of consciousness was found in 64 patients (67.4%) and headache in 18 (18.9%), together with neck stiffness in one case (Table 2). Sixteen patients (16.8%) experienced seizures, with multiple episodes in 11 cases and incoming in one. Less frequent manifestations were alterations of muscle tone (*n* = 4, 4.2%), balance disturbances (*n* = 3, 3.2 %), gait disorders (*n* = 3, 3.2%), nystagmus (*n* = 3, 3.2%), dizziness (*n* = 3, 3.2%), and photophobia (*n* = 3, 3.2%). Three patients (3.2%) presented anxiety; among them, one (1.1%) developed psychomotor agitation and another one had insomnia and factitious disorder. Other findings were motor deficits (*n* = 2, 2.1%), amnesia (*n* = 2, 2.1%), visual loss (*n* = 2, 2.1%), diplopia (*n* = 1, 1.1%), visual hallucinations (*n* = 1, 1.1%), and dysgeusia (*n* = 1, 1.1%). Three children (3.2%) presented only the neurologic involvement without other symptoms. 

#### 3.2.2. MIS-C Group

Neurological manifestations at onset were headache (*n* = 8, 14.8%) and impairment of consciousness (*n* = 2, 7.4%) (Appendix A). During the acute phase, 19 patients (70.4%) developed consciousness impairment including drowsiness, stupor, irritability, and disorientation (Table 2). Nine patients (33.3%) had headache and four (14.8%) meningism. Photophobia and phonophobia were complained in three cases (11.1%) and one (3.7%) case, respectively. Five patients (18.5%) presented mood deflection and five (18.5%) behavioral disorders such as psychomotor agitation, oppositional defiant disorder, fits of anger and apathy. Three patients (11.1%) had dysgeusia and two (7.4%) developed language alterations (slurred and poor speech) with preserved comprehension. Other neurological symptoms were dizziness (*n* = 2, 7.4 %), blurred vision (*n* = 1, 3.7%), anxiety (*n* = 1, 3.7%), and sleeping disorder (*n* = 1, 3.7%). 

#### 3.2.3. COVID-19 and MIS-C Comparisons

Neurological manifestations had longer latency in MIS-C than in COVID-19 (median time 2 [IQR 0–3] days vs. 0 [IQR 0–1] days, *p* 0.007) (Appendix A). At onset, impairment of consciousness was more frequent in the COVID-19 group (35.8% vs. 7.4%, *p* 0.004) while headache prevailed in the MIS-C one (29.6% vs. 12.6%, *p* 0.035). During the disease course, patients with COVID-19 presented more frequently with seizures (16% vs. 0%, *p* 0.021). Children with MIS-C had more often multiple neurological involvements (59.3% vs. 25.3%, *p* < 0.001) and specific manifestations such as consciousness reduction until stupor, confusion, mood and behavioral disorders, meningism, dysgeusia, and speech impairment.

### 3.3. Laboratory Tests

The main laboratory data are reported in Appendix A. In the MIS-C group, inflammatory markers, and white blood cells (WBC) count were statistically higher (*p* < 0.001), and lymphopenia, thrombocytopenia, and anemia were more frequent and severe (*p* < 0.001). In contrast, neutropenia was significantly more marked in the COVID-19 group (*p* < 0.001). All children underwent at least one SARS-CoV-2 test. In detail, 51 patients were tested with PCR alone, 60 underwent both tests, and 11 were diagnosed only by the antigenic test. Among patients who performed the molecular test (*n* = 111), 96 (78.7%) had a positive result, while among patients tested by antigenic method (*n* = 71), 55 (45.1%) resulted positive.

### 3.4. Instrumental Investigations

EEG was performed in nine patients (7.4%). A diffuse slow wave activity was found in two cases affected by MIS-C with severe consciousness impairment and in an infant with COVID-19 and multiple seizures followed by stupor. In the first two, EEG normalized at control. Furthermore, in an eight-year-old girl with three seizure episodes in 48 h, EEG showed diphasic intercritical abnormalities on the center-temporal regions bilaterally, clearly activated during sleep.

Brain CT scans, performed in 11 patients (9%), revealed alterations in two (1.6%). In the first case, a 12-year-old boy with headache, dizziness, diplopia, and ataxia, it revealed an extensive hypodensity of the left cerebellum and some hypodense spots in the thalami (Figure 2a); in the second one, a 10-month-old infant with MIS-C, with bulging anterior fontanelle and lethargy, it showed cerebral oedema and herniation.

Brain or spinal cord MRI, performed in 12 patients (9.8%), was altered in five (4.1%). In the above-mentioned 12-year-old boy, it showed an ischemic injury of the left cerebellum cortex (Figure 2b–d), and the left posterior inferior cerebellar artery (PICA) was unrecognizable in the angiographic sequences. Multiple thalamic ischemic injuries were found also in a 15-year-old boy with confusion and loss of consciousness. In the patient with GBS, the spinal cord MRI showed contrast enhancement of cauda equina roots. In the 10-month-old infant with MIS-C and cerebral oedema on CT, mild cerebral atrophy and areas of altered signal were revealed at MRI performed 2 weeks after the CT scan. In a two-year-old boy with MIS-C and impairment of consciousness, a second MRI, performed after one month from the clinical onset, showed diffuse cortical and subcortical brain atrophy. The three patients with stroke underwent cerebral DSA, which was normal in all but the boy with the cerebellar stroke. The instrumental data are shown in Table 3.

### 3.5. Management

Therapeutic strategies are reported in Appendix A. The MIS-C group patients needed therapeutic interventions more than COVID-19 ones (*p* < 0.001), except for benzodiazepines.

#### 3.5.1. COVID-19 Group

Steroids were orally administered in 10 cases (10.5%), and intravenously (IV) in six (6.3%). Three patients (3.2%) received IV immunoglobulins (IVIg): two cases with an arterial ischemic stroke (AIS) received a 2 g/kg single dose and the patient with GBS 400 mg/kg/die for five days. Benzodiazepines were administered in eight (8.4%) children for seizures (*n* = 6) or psychomotor agitation (*n* = 2). Patients with previous neurological comorbidities continued their habitual therapy, but in three cases anti-seizure medications were increased. In the child with a new diagnosis of epilepsy, anti-seizures therapy was introduced.

#### 3.5.2. MIS-C Group

Steroids were administered by IV in 24 (88.9%) patients and orally in 23 (85.2%). All patients received one dose of IVIg (2 g/kg), but four required a second dose. An anti-interleukin-1 receptor antagonist was administered in 13 children (48.1%). Heparin was administered in 12 (44.4%) patients by IV or subcutaneously. The child with cerebral oedema was treated with mannitol and hypertonic solution. Benzodiazepines were successfully administered in one case (3.7%) for controlling a psychomotor agitation.

### 3.6. Course and Outcome

Course and outcome data are reported in Appendix A. In our cohort, no patient died. Admission to the Intensive Care Unit (ICU) resulted more frequently in the MIS-C group (48.1% vs. 3.2%, *p* < 0.001). Such admission was due to respiratory distress in three patients (3.2%) affected by COVID-19, and to hemodynamic failure in those with MIS-C, but in three cases a severe neurological involvement was also present. The length of stay was longer in the MIS-C group than in the COVID-19 group, whereas the persistence of sign or symptoms at discharge or last follow-up were not significantly different in the two groups. Among a total of 86 patients (70.5%) who underwent follow-up, after 14–497 days from discharge (median 122 [IQR 71.5–197.5] days), only six (4.9%) had persistent neurological impairment: confusion, anxiety, osteotendinous reflexes reduction, and vision loss in the COVID-19 group, dysgeusia and attention deficit in the MIS-C group.

### 3.7. Specific Neurological Disorders

Three patients were diagnosed with AIS during the COVID-19. A 12-year-old boy had a left cerebellar stroke (Figure 2) due to PICA occlusion with a complete clinical recovery but partial radiological improvement. The second was a 15-year-old boy with consciousness impairment and bilateral thalamic ischemic lesions. Despite a clinical improvement, a one-month control brain MRI showed a new cortex ischemic injury. At discharge, his neurological impairments improved with persistent confusion episodes. The third was a six-year-old girl who presented a right central retinal artery occlusion (CRAO) together with an idiopathic contralateral optic neuropathy (Figure 3). Neuroimaging (brain CT, MRI, and angiography) was normal. She completely lost her right eye function, but she recovered her fingers counting on the left.

A two-year-old boy with SARS-CoV-2 infection and progressive sensitive and motor limbs impairment was diagnosed with GBS by clinical manifestations, electroneurography, liquor and MRI findings. At the six-months follow-up, he showed just mild lower limb osteotendinous reflexes reduction with no other sequelae.

An eight-year-old girl with COVID-19 was newly diagnosed with childhood epilepsy with centro-temporal spikes (CECTS) based on clinical and EEG alterations.

Eleven children were diagnosed with febrile seizures (first episode in nine of them), and complex seizures have occurred in seven patients (focal symptoms (*n* = 2) or recurrent episodes (*n* = 6)). Three patients, one with COVID-19 and two with MIS-C, were diagnosed with encephalopathy based on clinical and EEG findings; lumbar puncture was not performed in these patients.

## 4. Discussion

Neurological involvement is well known even at pediatric age, both during COVID-19 and MIS-C, consisting of a heterogeneous clinical spectrum [16,17]. In a large case series of 1695 children, 12% with neurological manifestations had a life-threatening course [26].

In our study, neurological symptoms or signs were reported in about 66% of patients with MIS-C and 31% with COVID-19. The latter percentage might not reflect the real incidence of neurological involvement because not all COVID-19 patients are hospitalized. The incidence of neurological involvement is highly variable in the literature, ranging from 3.8% to 67% [16,17,18,26]. Furthermore, some authors did not report any type of neurological manifestations [30]. This variability might be explained by study design, the frequency of nasopharyngeal swab performance in hospitalized patients, and even the different SARS-CoV-2 variants succession. Differences between different SARS-CoV-2 diagnostic tests in our cohort depended on the time of patient admission. Nowadays, the PCR test still represents the reference target to reach safe conclusions; otherwise, some of the available third-generation antigen tests have very high sensitivity and specificity comparable with the PCR test.

The selection of our patients included any neurological symptom, including mild and transient consciousness impairment, these manifestations being often responsible for Emergency Department (ED) admission and hospitalization.

Although the pathogenesis of neurological involvement due to SARS-CoV-2 is not completely understood, inflammatory reaction, post-infectious auto-immune response, brain damage secondary to organs dysfunction, and viral direct neurotropic effect have been described [4,5,10,31]. The latter occurrence is supported by SARS-CoV-2 findings in the cerebral parenchyma at post-mortem examination and in the cerebrospinal fluid (CSF) during meningitis, encephalitis or GBS [32,33,34,35,36]. According to our study, the viral detection on the CSF is rare, as confirmed by Lewis et al. in a literature review, but specific research on the viral genome is not commonly performed [37,38]. Another fundamental mechanism is the prothrombotic effect on the nervous system vessels, with the induction of a procoagulant state and the coagulative cascade activation [5,10,39]. In MIS-C, most of the features seem due to an altered immune-mediated response [16,25].

The Global Consortium Study of Neurologic Dysfunction in COVID-19 recently identified potential risk factors for neurologic involvement in patients with COVID-19 or MIS-C [17]. Even in our series, neurological manifestations were more frequent during the MIS-C course, and both groups referred with some frequency pre-existing neurological conditions, already related to an increased risk of neurological involvement in other studies on adults and children [13,26]. In contrast, our population had an early median age, being younger compared to other pediatric series [16,18,26]. Neurological symptoms often occurred early in patients affected by COVID-19, whereas in the MIS-C group they developed after a longer latency period, in agreement with Ray et al., and consistently with a different pathogenetic mechanism [16].

In our series, neurological involvement ranged in a wide spectrum of manifestations in agreement with the literature. In the COVID-19 group, the most frequent manifestations were impairment of consciousness and headache, as in most of the previous studies. Otherwise, in a systematic revision of the literature headache had a prevalence of just 3% [40]. In the COVID-19 group, seizures were experienced by about 16% of patients, more frequently than in other cohorts [18,41]. This difference may be related to a history of neurologic comorbidities in one-fifth of our patients, with a possible increased risk of convulsions, as reported by Kurd et al. [41]. In addition, the different timing of the studies may play an important role. Ludvigsson et al. described an increased incidence of seizures during the Omicron wave [42]; in our study, none of the patients with convulsions underwent viral genome analysis, but almost all of them (15 of 16) manifested the episode between January and June 2022, when the Omicron variant was the most frequent in Italy [43]. In our series, even children younger than two years experienced seizures, unlike another observational study [18]; however, the incidence of convulsions was similar in the under-three and three-ten years age groups. In the MIS-C group, about 70% of patients experienced consciousness impairment, about 20% behavioral changes, and another 20% mood deflection, which have been rarely mentioned [25]. Many studies reported drowsiness, irritability, and confusion as the most frequent manifestations of MIS-C [16,24,25,26,44]. Olivotto et al. recently described two phenotypes of encephalopathy during the MIS-C course: a mild form, characterized by irritability, drowsiness, mood deflection, and headache, and a severe one, characterized by meningism, photophobia, and focal CNS involvement [25]. In our study, four and three patients, respectively, had meningism and photophobia, and two patients experienced a peculiar speech impairment, as described by the above-mentioned study [25]. Headache, one of the most frequent symptoms among children with MIS-C, reported up to 100% in a single-center case series [25], was referred by about one-third of our patients. Children affected by MIS-C more often presented multiple neurological manifestations compared to patients with COVID-19. Consciousness impairment was the symptom most observed in both groups, and the MIS-C group more frequently experienced consciousness decline such as hypo-reactivity, drowsiness, confusion, lethargy until stupor, associated with behavioral changes. Acute encephalopathy more commonly affected patients with MIS-C in a multinational study cohort [16]. Several other manifestations, such as speech impairment, meningism and dysgeusia, were also more frequently observed in the MIS-C group, as opposed to seizures. This diversity may be partially influenced by the different groups’ ages. Few studies analyzed the variability of neurological presentation according to age [18,26]. In our series, in agreement with Riva et al. [18], consciousness impairment and seizures were more common in the younger patients, whereas headache, dizziness, and photophobia in the older ones; even dysgeusia, behavioral changes, or other psychiatric disorders were not reported in children younger than three years old. This reflects the difficulty of the youngest to express some neurological symptoms. Likewise, smell and taste alterations are more rarely reported in children compared to adults [13,14,45].

Some of our patients experienced psychiatric symptoms, mostly in the MIS-C group. Psychiatric disorders such as psychosis, depression, mania, and anxiety have been reported in adults affected by COVID-19 [14,15]. Otherwise, the literature rarely reports psychiatric symptoms in children with COVID-19 and even less with MIS-C [16,17,29]. The role of the infection as a trigger is not completely clear. Multiple mechanisms, mostly on an inflammatory basis, have been described and anti-SARS-CoV-2 antibodies were found on the CSF of two adolescents with COVID-19 and subacute neuropsychiatric symptoms [15,29]. In addition, the pandemic has had a significant impact on mental health since the beginning, even indirectly [46]. Therefore, in our opinion, psychiatric symptoms should be looked for in patients with SARS-CoV-2 infection, and a neuropsychiatric evaluation should be requested in the suspicious cases for a prompt diagnosis and adequate management.

Although uncommon, stroke is a possible complication of COVID-19 and MIS-C, even in the pediatric age, and several cases are reported in the literature [16,17,26,27,47]. In this series, three patients experienced an ischemic stroke during the COVID-19 course, and one of them specifically presented a retinal infarction due to CRAO. [48]. To the best of our knowledge, our case is the first report of CRAO in children with SARS-CoV-2 infection. Nevertheless, several cases have been reported in adults, and a combined central retinal artery and vein occlusion has been described in a 20-year-old Indian [49,50,51]. In addition, it is known that the SARS-CoV-2 infection prothrombotic effect may also involve the ocular district [52]. The girl in our study with CRAO experienced a contralateral idiopathic optic neuropathy. Optic neuritis has been reported during COVID-19, as well as other viral infections; however, in our case, brain and orbits MRI excluded optic nerve signal alterations [16,27].

Guillain-Barré syndrome with classic clinical, CSF and radiological alterations was diagnosed in one child. The association between SARS-CoV-2 infection and GBS is well-known in the literature, and about hundred and twenty cases have been reported, respectively, in the adult and pediatric populations [53,54]. Our patient had a shorter latency between systemic symptoms and neurological disease onset compared to the median latency reported by a systematic review [53], but these authors highlights even earlier neurological manifestations. In a UK prospective national study, a considerable group of patients was recognized to be affected by neuroimmune disorders, mostly in the COVID-19 group, but also after acute infection healing [16]. An immune-mediated pattern resulted even the most frequent neuroimaging findings [55]. In our series, only one child developed a neuroimmune condition (GBS).

Encephalitis or aseptic meningitis, although reported in several studies [5,17,26,28], was not found in our series. This could be due to the low rate of neurological assessments such as rachicentesis, EEG, and radiologic tests in our cohort compared to others [16].

Encephalopathy is a known complication of SARS-CoV-2 infection [16,25,26], as already reported for other viruses such as influenza [56]. Only three of our patients received this diagnosis, presenting evocative signs and reversible EEG abnormalities, like those described by other studies [25]. Nevertheless, similarly to other CNS inflammations, the encephalopathy incidence could be underestimated in our series, considering the high frequency of the key, suggestive symptoms.

All the MIS-C patients had systemic involvement, according to diagnostic criteria [19,57]. Almost all COVID-19 patients experienced extra-neurological unspecific manifestations but, in a few cases, isolated neurological symptoms were reported. Thus, it is advisable to investigate SARS-CoV-2 infection in all children with acute neurologic manifestations [16,27].

Neuroimaging abnormalities were uncommon in our study, although a brain MRI was performed in only 10% of patients, similar to other studies [18,28]. Heterogeneous neuroimaging findings are reported in the literature, and Lindan et al. described various imaging patterns, even during asymptomatic SARS-CoV-2 infection [26,55]. Bilateral thalamic lesions and reversible splenial lesions have been documented in patients with encephalopathy/encephalitis [16,24,44]. These findings were not identified in our patients. However, diffuse cortical/subcortical atrophy was described in two cases, other than bilateral thalamic ischemic lesions in one of the three stroke COVID-19 patients. Even in the setting of COVID-19 and MIS-C, very few cases of AIS have been reported in the literature. Beslow et al. found an incidence of 0.82% (*n* = 8) of AIS in pediatric patients with SARS-CoV-2 (*n* = 6) and MIS-C (*n* = 2), with medium cerebral artery (MCA) as the most frequent vessel involved [47]. In our series, the incidence of AIS was about 0.87% with involvement of PICA in one case and CRAO in another one, and no cases of AIS were found in MIS-C group. Steroid treatment was deemed a plausible cause of brain atrophy, although outcomes of brain damage or inflammation cannot be excluded, as already reported in patients with a history of brain injury, systemic chronic inflammatory disorders, and autoimmune encephalitis, even at pediatric age [58,59,60].

Reports of severe or life-threatening neurologic manifestations are growing in the literature, and neurological involvement was ascribed as a risk factor for mortality, ICU admission, longer hospital and ICU stay, and higher need for rehabilitation [17,25,26]. None of our patients died, but 13% of them required ICU admission, due to neurological deterioration in two cases of MIS-C.

Neurological manifestations were mostly transient, in agreement with other papers [17,26]; however, 13% and 5% of children still presented with neuropsychiatric symptoms or signs, respectively, at hospital discharge and last follow-up, and five patients showed abnormalities at the final neuroimaging. The neurologic outcome resulted similarly in both groups, in line with Ray et al. [16]. It is noteworthy that the impact on cognitive abilities and neuropsychic development may be underestimated in the absence of standardized neuropsychological tests administration and a long-term follow-up.

This study has some limitations. Data were available only when reported in the medical record, due to the retrospective design. The enrolled patients did not always undergo specialistic neurologic evaluations, and symptoms and signs may have been poor recognized or unclearly framed. The study was conducted in a single pediatric tertiary-level center, and this may have influenced the characteristics of the cohort, presumably different from the general pediatric population. In our series, about 20% of patients referred pre-existing neurological or psychiatric comorbidities, whereas other children were affected by concomitant infections; the role of these conditions on the clinical course is unpredictable. Finally, in the COVID-19 group, only patients with acute SARS-CoV-2 infection were included, missing children who developed neurological disorders following the infection, including long-COVID clinical features [61].

## 5. Conclusions

The present study provides a comprehensive review of COVID-19- related neurological manifestations in children. In clinical practice, these reports underline the need to make an accurate patient history and neurological examination in all children with COVID-19 or MIS-C. On the other hand, during the ongoing pandemic, all children presenting with neurological manifestations should be checked for SARS-CoV-2 infection or MIS-C.

Neurological involvement is frequent in the pediatric age, in patients affected by both COVID-19 and, even more, MIS-C. Neurological symptoms and signs are highly heterogeneous. The frequency of certain neurological manifestations varies depending on the patient’s age. Symptoms and signs can even be differentiated in the two groups, COVID-19 and MIS-C, but consciousness impairment is the most frequent manifestation in both groups, and may underlie an encephalopathy. The latency of neurologic manifestations was greater in the MIS-C group, so suspicion should remain high in the days after the onset. Most neurological manifestations were mild in our series; however, severe complications, such as ischemic stroke and GBS, are worthy of note. Other diagnoses, such as encephalopathy, encephalitis, and meningitis, were rare or absent in our series, but neurologic assessments were performed with a low frequency. A diagnostic pathway implementation is suggested in these patients, to improve the detection of the disorder and to optimize the treatment. We underline the importance of considering psychiatric symptoms during both COVID-19 and MIS-C and recommend a neuropsychiatric evaluation in suspicious cases. 

A large part of the manifestations were transient, and clinical or radiological findings persisted at follow-up in a minority of children; however, the eventual impact on neurocognitive abilities and psychic equilibrium may be underestimated.

## Figures and Tables

**Figure 1 children-09-01809-f001:**
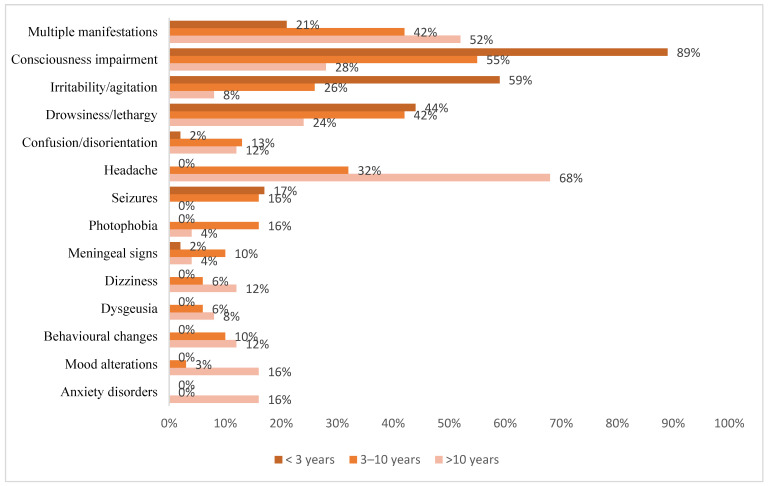
Stratification by age of main neurological manifestations.

**Figure 2 children-09-01809-f002:**
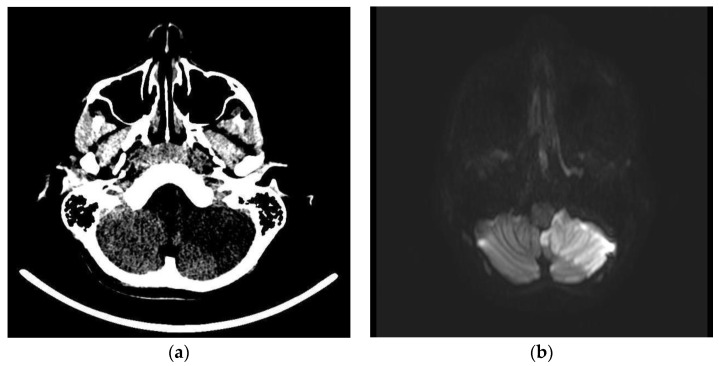
Acute left cerebellar infarct in the distribution of the posterior inferior cerebellar artery (PICA) without mass effect and enhancement areas. Axial brain CT scan (**a**) and axial brain MRI images of DWI (**b**), ADC (**c**), and T1- WI after intravenous gadolinium contrast administration (**d**) sequences.

**Figure 3 children-09-01809-f003:**
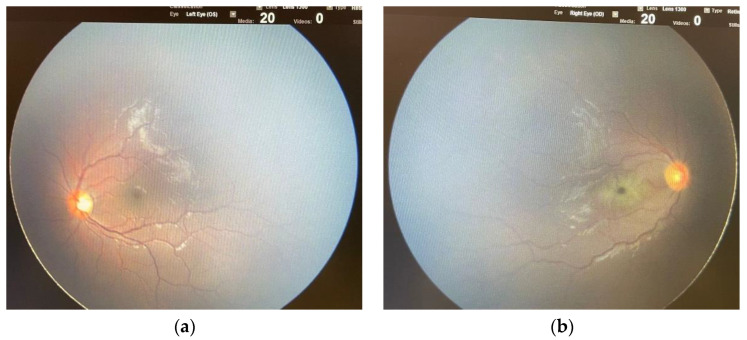
Fundus examination in a six-year-old girl with central retinal artery occlusion (CRAO) during COVID-19 (with kind concession of P. Fortunato MD). (**a**) Normal fundus in the left eye. (**b**) In the right eye characteristic funduscopic findings of CRAO: retinal whitening, cherry red spot in the fovea and filiform retinal arteries.

**Table 1 children-09-01809-t001:** Demographic data in the enrolled 122 patients and in the two subgroups.

Epidemiological Data	Total (*n* = 122)	COVID-19 (*n* = 95)	MIS-C (*n* = 27)	*p*
Age at onset, median (IQR)—years	1.98 (0.34–8.57)	0.94 (0.21–6.98)	8.5 (3.68–11.62)	<0.001
Gender: Male, *n* (%)	74 (60.7)	53 (55.8)	21 (77.8)	0.039
Ethnicity: Caucasian, *n* (%)	101 (82.8)	81 (85.3)	20 (74.1)	0.227
Others, *n* (%)	21 (17.2)	14 (14.7)	7 (25.9)	
Asiatic, *n*	5	2	3	
Hispanic, *n*	4	4	0	
African, *n*	3	0	3	
Pacific Islands, *n*	1	1	0	
Not specified, *n*	8	7	1	
Familiar history ^#^, *n* (%)	23 (18.9)	20 (21.1)	3 (11.1)	0.402
Comorbidities, *n* (%)	53 (43.4)	42 (44.2)	11 (40.7)	0.748
Neuropsychiatric comorbidities, *n* (%)	24 (19.7)	21 (22.1)	3 (11.1)	0.277
Time between onset and hospitalization, median (IQR)—days	3 (1–5)	2 (1–5)	4 (3–6)	<0.001

^#^ Positive family history was intended for neuropsychiatric disorders, autoimmune conditions and thrombotic events.

**Table 2 children-09-01809-t002:** Neurological manifestations in the whole disease course.

Neurological Manifestations	Total (*n* = 122)	COVID-19 (*n* = 95)	MIS-C (*n* = 27)	*p*
Multiple manifestations, *n* (%)	40 (32.8)	24 (25.3)	16 (59.3)	<0.001
Consciousness impairment, *n* (%)	83 (68)	64 (67.4)	19 (70.4)	0.822
Irritability/agitation, *n* (%)	49 (40.2)	37 (38.9)	12 (44.4)	0.607
Drowsiness/hyporeactivity, *n* (%)	48 (39.3)	32 (33.7)	16 (59.3)	0.016
Confusion, *n* (%)	8 (6.6)	3 (3.2)	5 (18.5)	0.013
Temporary LOC, *n* (%)	2 (1.6)	2 (2.1)	0	1
Sopor/stupor, *n* (%)	3 (2.5)	0	3 (11.1)	0.01
Headache, *n* (%)	27 (22.1)	18 (18.9)	9 (33.3)	0.112
Seizures, *n* (%)	16 (13.1)	16 (16.8)	0	0.021
Behavioural changes, *n* (%)	6 (4.9)	1 (1.1)	5 (18.5)	0.002
Mood disorders, *n* (%)	5 (4.1)	0	5 (18.5)	<0.001
Anxiety disorders, *n* (%)	4 (3.3)	3 (3.2)	1 (3.7)	1
Photophobia, *n* (%)	6 (4.9)	3 (3.2)	3 (11.1)	0.122
Phonophobia, *n* (%)	1 (0.8)	0	1 (3.7)	N.A.
Meningeal signs, *n* (%)	5 (4.1)	1 (1.1)	4 (14.8)	0.008
Bulging fontanelle, *n* (%)	2 (1.6)	1 (1.1)	1 (3.7)	0.395
Dizziness, *n* (%)	5 (4.1)	3 (3.2)	2 (7.4)	0.306
Dysgeusia/ageusia, *n* (%)	4 (3.3)	1 (1.1)	3 (11.1)	0.034
Hyper/hypotonia, *n* (%)	4 (3.3)	4 (4.2)	0	0.575
Balance deficit, *n* (%)	3 (2.5)	3 (3.2)	0	1
Gait alterations, *n* (%)	3 (2.5)	3 (3.2)	0	1
Motor deficit, *n* (%)	2 (1.6)	2 (2.1)	0	1
Retrograde amnesia, *n* (%)	2 (1.6)	2 (2.1)	0	1
Speech disturbances, *n* (%)	2 (1.6)	0	2 (7.4)	0.048
Visual hallucinations, *n* (%)	1 (0.8)	1 (1.1)	0	N.A.
Visual impairment, *n* (%)	3 (2.5)	2 (2.1)	1 (3.7)	0.531
Nystagmus, *n* (%)	3 (2.5)	3 (3.2)	0	1
Strabismus, *n* (%)	1 (0.8)	1 (1.1)	0	N.A.
Double vision, *n* (%)	1 (0.8)	1 (1.1)	0	N.A.
Others: Sleeping disorders, *n* (%)	2 (1.6)	1 (1.1)	1 (3.7)	0.395
Factitious disorder, *n* (%)	1 (0.8)	1 (1.1)	0	N.A.
Neuropathic pain, *n* (%)	1 (0.8)	1 (1.1)	0	N.A.
Osteotendineous reflexes deficit, *n* (%)	1 (0.8)	1 (1.1)	0	N.A.

LOC: loss of consciousness; N.A.: not applicable.

**Table 3 children-09-01809-t003:** Altered instrumental and radiological examinations.

Tests	Total (*n* = 122)	COVID-19 (*n* = 95)	MIS-C (*n* = 27)	*p*
EEG, *n* (%)	4 (3.3)	2 (2.1) Diffuse symmetric slow wave activity (1) centro-temporal spikes (BRE/CECTS) (1)	2 (7.4) Diffuse symmetric slow wave activity (2)	0.212
ENG/EMG, *n* (%)	1 (0.8)	1 (1.1) NCV slowing, absent F waves	0	N.A
Flash VEP, *n* (%)	1 (0.8)	1 (1.1) Left optic NCV impairment	0	N.A
TCS/TCD, *n* (%)	0	0	0	N.A.
Brain CT, *n* (%)	2 (1.6)	1 (1.1) Parenchimal hypodensity	1 (3.7) Brain oedema and herniation	0.395
Brain/spinal cord MRI, *n* (%)	5 (4.1)	3 (3.2) Brain ischemic lesions (2), cerebral artery occlusion (1), contrast enhancement cauda equina (1)	2 (7.4) Cranial nerve contrast enhancement (1) cortical/subcortical atrophy (2), periventricular/ peritrigonal hyperintense signal (1)	0.306
DSA, *n* (%)	1 (0.8)	1 (1.1) Partial left PICA occlusion	0	N.A.

BRE: Benign Rolandic Epilepsy; CECTS: childhood epilepsy with centro-temporal spikes; CT: computed tomography; DSA: digital subtraction angiography; EEG: electroencephalogram; EMG: electromyography; ENG: electroneurography; MRI: magnetic resonance imaging; N.A.: not applicable; NCV: nerve conduction velocity; PICA: posterior inferior cerebellar artery; TCS: transcranial sonography; TCD: transcranial Doppler ultrasound; VEP: visual evoked potentials.

## Data Availability

The data presented in this study are available on request from the corresponding author.

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
