# Peer review of "Neurological Involvement in Children with COVID-19 and MIS-C: A Retrospective Study Conducted for More than Two Years in a Pediatric Hospital"

_children, 2022, doi:10.3390/children9121809_

Round 1
Reviewer 1 Report
1. The paper is well written and interesting to read however, I couldn't find the supplement tables.
2. In order to be able to arrive to safe conclusions, do you believe that antigen positive samples to SARS - CoV-2 should also be confirmed by PCR?
to be able to draw firm conclusions, do you think antigen positive samples should also be confirmed by pcr?
Author Response
Comment 1: Sorry for the inconvenience. Supplement tables have been attached to the manuscript in the first submission. However, we newly attached the file with the revised manuscript.
Comment 2: Thanks for the suggestion with which we fully agree. We consider the PCR test as the reference target; however, the actual validated third-generation antigenic test should be considered a comparable alternative method. We enriched the text of the manuscript by adding a few sentences accordingly to your suggestion in the Discussion (lines 285-289).
The differences in the SARS-CoV-2 tests in our series depended on the time of the patient's admission. At the outbreak’s beginning, the only available test was the molecular one. When the third-generation antigenic test became available in our lab, our hospital’s SARS-CoV-2 protocol changed and both antigenic and PCR tests were performed at admission. After analysing one-year experience, our immunologists found that the sensitivity and specificity of the third-generation antigenic test were overlapping with PCR. Subsequently, we used the antigenic test as the only SARS-CoV-2 diagnostic method; the molecular one was reserved for the borderline antigenic test. In detail, 51 patients were tested with PCR alone, 60 underwent both tests, and 11 were diagnosed only by the antigenic test. This clarification has been added in the Results section (Laboratory test paragraph)
Reviewer 2 Report
countless retrospective reports on Covid and its association with other diseases have been made, it would be convenient to determine the possible impact of these reports on clinical practice and in particular this report
Author Response
Thanks for the suggestion with which we fully agree. As you correctly underlined, innumerable reports on the association between COVID-19 and other diseases have been published and a lot of them are focused on neurological involvement. In this regard, the present study provides a comprehensive review of COVID-19- related neurological manifestations in children. In clinical practice, these reports underlined the need to make an accurate patient history and neurological examination in all children with COVID-19 or MIS-C. On the other hand, during the ongoing pandemic, all children presenting with neurological manifestations should be checked for SARS-CoV-2 infection or MIS-C. Accordingly, we integrate the text of the paper adding such comments in the Conclusions section (lines 444-450).
Reviewer 3 Report
In Fig 2, images from different views are used. It is recommended to use images from the same area and the same view. And it would be good to add DWI as well.
Author Response
Thanks for the suggestion with which we fully agree. We modified images of figure 2 as you suggested, selecting axial sections at the same level, in ADC, T1 weighted and DWI sequences.